# Retrospective *in silico* mutation profiling of SARS-CoV-2 structural proteins circulating in Uganda by July 2021: Towards refinement of COVID-19 disease vaccines, diagnostics, and therapeutics

Steven Odongo[1,2,3]*, Hedmon Okella[2,4], Christian Ndekezi[5], Moses Okee[6,7], Monica Namayanja[2], Brian Mujuni[7], Yann G. J. Sterckx[8], Dennison Kizito[5], Magdalena Radwanska[3,9], Stefan Magez[3,10], Kokas Ikwap[11], Frank Nobert Mwiine[11], Julius Julian Lutwama[5], Charles Ibingira[12]

1 Department of Biotechnical and Diagnostic Sciences, College of Veterinary Medicine, Animal Resources and Biosecurity (COVAB), Makerere University, Kampala, Uganda, 2 Center for Biosecurity and Global Health, College of Veterinary Medicine, Animal Resources and Biosecurity (COVAB), Makerere University, Kampala, Uganda, 3 Laboratory for Biomedical Research, Ghent University Global Campus, Incheon, South Korea, 4 Pharm-Biotechnology and Traditional Medicine Center, Mbarara University of Science and Technology, Mbarara, Uganda, 5 Uganda Virus Research Institute, Entebbe, Uganda, 6 Department of Medical Microbiology, College of Health Sciences, Makerere University, Kampala, Uganda, 7 Africa Center of Excellence in Materials, Product Development and Nanotechnology, College of Engineering, Makerere University, Kampala, Uganda, 8 Laboratory of Medical Biochemistry and the Infla-Med Centre of Excellence, University of Antwerp, Campus Drie Eiken, Universiteitsplein, Wilrijk, Belgium, 9 Department of Biomedical Molecular Biology, Ghent University, Ghent, Belgium, 10 Laboratory of Cellular and Molecular Immunology, Vrije Universiteit Brussel, Brussels, Belgium, 11 Department of Biomolecular Resources, College of Veterinary Medicine, Animal Resources and Biosecurity, Makerere University, Kampala, Uganda, 12 Department of Anatomy, School of Biomedical Sciences, College of Health Sciences, Makerere University, Kampala, Uganda

* steven.odongo@mak.ac.ug

## Abstract

The SARS-CoV-2 virus, the agent of COVID-19, caused unprecedented loss of lives and economic decline worldwide. Although the introduction of public health measures, vaccines, diagnostics, and therapeutics disrupted the spread of the SARS-CoV-2, the emergence of variants poses substantial threat. This study traced SARS-CoV-2 variants circulating in Uganda by July 2021 to inform the necessity for refinement of the intervention medical products. A comprehensive *in silico* analysis of the SARS-CoV-2 genomes detected in clinical samples collected from COVID-19 patients in Uganda revealed occurrence of structural protein variants with potential of escaping detection, resisting antibody therapy, or increased infectivity. The genome sequence dataset was retrieved from the GISAID database and the open reading frame encoding the spike, envelope, membrane, or nucleocapsid proteins was translated. The obtained protein sequences were aligned and inspected for existence of variants. The variant positions on each of the four alignment sets were mapped on predicted epitopes as well as the 3D structures. Additionally, sequences within each of the sets were clustered by family. A phylogenetic tree was constructed to assess relationship between the encountered spike protein sequences and Wuhan-Hu-1 wild-type, or the

**Data Availability Statement:** All relevant data are within the manuscript and its Supporting Information files.

**Funding:** S.O. received an award for conducting this study from the Government of Uganda through the Makerere University Research and Innovation Fund (grant number MAK-RIFDVCFA/026/20) at https://rif.mak.ac.ug/. The funders had no role in study design, data collection and analysis, decision to publish, or preparation of the manuscript.

**Competing interests:** The authors have declared that no competing interests exist.

*Alpha*, *Beta*, *Delta* and *Gamma* variants of concern. Strikingly, the frequency of each of the spike protein point mutations F157L/Del, D614G and P681H/R was over 50%. The furin and the transmembrane serine protease 2 cleavage sites were unaffected by mutation. Whereas the *Delta* dominated the spike sequences (16.5%, 91/550), *Gamma* was not detected. The envelope protein was the most conserved with 96.3% (525/545) sequences being wild-type followed by membrane at 68.4% (397/580). Although the nucleocapsid protein sequences varied, the variant residue positions were less concentrated at the RNA binding domains. The dominant nucleocapsid sequence variant was S202N (34.5%, 205/595). These findings offer baseline information required for refining the existing COVID-19 vaccines, diagnostics, and therapeutics.

## Introduction

Severe acute respiratory syndrome coronavirus strain 2 (SARS-CoV-2) caused the outbreak of the coronavirus disease 2019 (COVID-19) pandemic [1, 2], which has affected millions of lives around the world, and continues to cause deaths nearly three years after the emergence of the disease. The World Health Organization (WHO) estimated confirmed cumulative cases and deaths, as of the 2nd September 2022, at 601,189,435 and 6,475,3346, respectively; with new cases per day at 618,970 [3]. Global excess deaths associated with COVID-19 for the period January 2020 –December 2021 were estimated at 14.91 million [4]. According to the World Bank [5], COVID-19 has affected 1.6 billion workers so far, especially in the wholesale and retail businesses, food and hospitality, tourism, transport and manufacturing industries. Key interventions, which reduced COVID-19 hospitalization and deaths were public health measures [6], COVID-19 tests [7], vaccines [8] and therapeutics [9]. Sustainability of this achievement will be guaranteed by conducting constant surveillance as well as evaluating performance of the existing interventions.

The SARS-CoV-2 virus belongs to lineage 2b of the genus *Betacoronavirus* and the subgenus *Sarbecovirus* [2, 10]. The genome of this virus shares 87.99% identity with the bat SARS-like CoV (bat-SL-CoVZC45, MG772933.1) [2]; however, the new virus was named 2019-new coronavirus (2019-nCoV) by the WHO [11] because the identity of its conserved replicase domains (ORF1ab) is less than 90% of the beta coronavirus. Six other human coronaviruses (HCoVs) are HCoV-HKU1, HCoV-MERS, HCoV-SARS-CoV, HCoV-229E, HCoV-OC43 and HCoV-NL63. Whereas HCoV-HKU1, HCoV-MERS, and HCoV-SARS-CoV belong to the genus *Betacoronavirus*, HCoV-229E, HCoV-OC43 and HCoV-NL63 belong to the genus *Alphacoronavirus* [12].

The morphology of SARS-CoV-2 is that of a spherical virus decorated with spike, envelope and membrane structural proteins traversing the virus envelope. Encased within the virus core is a 26 to 32 kb linear positive-sense, single-stranded RNA genome tightly bound by nucleocapsid protein. The genome of SARS-CoV-2 has 5'-cap structure, 17 open reading frames (ORFs) (1a, 1b, S, 3a, 3c, 3d, 3b, E, M, 6, 7a, 7b, 8, N, 9b, 9c and 14) and a 3' poly adenine tail [13]. The 5'- cap structure protects the genome from degradation by the host cytoplasmic endonucleases. ORF 1ab is translated into a polyprotein, which is proteolytically cleaved into 16 non-structural proteins (replicase complex). Subsequent ORFs are individually transcribed into sub-genomic RNAs prior to translation [14]. Aware of the diverse roles of SARS-CoV-2 proteins reviewed by Yadav et al. [13], we only highlighted the roles of structural proteins are herein. The spike protein attaches the virus particle onto a cell surface receptor angiotensin-

converting enzyme 2 (ACE2) [15] and its cleavage by furin as well as transmembrane serine protease 2 (TMPRSS2) are essential for proteolytic activation of SARS-CoV-2 permitting host cell entry [16]. The envelope protein forms a transmembrane ion channels [17], membrane protein is required for virus assembly [18], and nucleocapsid protein protects the virus genome [17]. Although membrane protein is the most abundant virus structural protein [13], Poran et al. [19] showed that nucleocapsid protein is the most abundant protein inside SARS-CoV-2 infected host cells.

SARS-CoV-2 proteins form the basis of the existing COVID-19 vaccines, immunoassays, and immunotherapies. Indeed, vaccine developments are constructed around 11 protein engineering platforms namely protein subunit, non-replicating Viral Vector (VVnr), DNA, Inactivated Virus, RNA, replicating Viral Vector (VVr), Virus Like Particle, VVr and Antigen Presenting Cell combination, Live Attenuated Virus, VVnr and Antigen Presenting Cell combination, and Bacterial antigen-spore expression vector. Viral protein(s) engineered through these platforms either evoke neutralizing antibodies or induce CD4 and CD8-cell responses. Heinz and Stiasny [20] comprehensively reviewed mechanisms of first-generation COVID-19 vaccines. COVID-19 immunoassays such as Panbio™ (Abbott), Standard Q COVID-19 Ag Home Test (SD Biosensor) and Biocredit COVID-19 Ag Home Test Nasal (BioVendor R&D™) are some of the commercial tests targeting SARS-CoV-2 proteins. Monoclonal antibody products targeting SARS-CoV-2 spike proteins are also in use [21].

Evidence shows that emerging SARS-CoV-2 variants [22] undermine vaccines [23, 24], escape detection by diagnostic tests [25–27], or resist antibody therapies [24, 28–30]. Variants are classified into lineages/families based on their observed similarity in amino acid substitution/deletion at the same mutation site(s) [31, 32]. Furthermore, classification is based on specific attributes such as resulting public health action, changes to receptor binding, reduced neutralization by antibodies generated against previous infection or vaccination, reduced treatment efficacy, potential diagnostic impact or predicted increase in transmissibility or disease severity. These include variants of concern (VOC), Variants of High Consequence (VOHC), Variants of Interest (VOI) and Variants Being Monitored (VBM) [22]. Note that first-generation COVID-19 medical products losing performance against emerging variants were developed based on information acquired from Wuhan-Hu-1 reference strain (NC_045512.2) genome. Therefore, surveillance of SARS-CoV-2 mutation profiles would unequivocally inform the refinement of the existing COVID-19 medical products to effectively tackle challenges imposed by these emerging variants. As such, the WHO advocates for "*continual assessment of genomic diversity, including in antigenically important sites that may be under selection, to help identify plausible candidate sites that might affect the efficacy of serological assays*" [33]. To this effect, a comprehensive retrospective analysis of the heterogeneity of SARS-CoV-2 structural proteins derived from genome sequences originating from Uganda was conducted.

## Materials and methods

### Dataset retrieval

SARS-CoV-2 genome sequences (n = 600) were retrieved from the Global Initiative on Sharing All Influenza Data (GISAID) database [34]. The downloaded sequences were those deposited by Medical Research Council/Uganda Virus Research Institute, London School of Hygiene & Tropical Medicine Uganda Research Unit, Entebbe (MRC/UVRI & LSHTM) (n = 572), and Makerere University College of Health Sciences (MAKCHS) (n = 28), and represented the totality of the then available sequences submitted to GISAID from Uganda. These sequences were derived from samples collected from the 21st March 2020, when the first case of COVID-

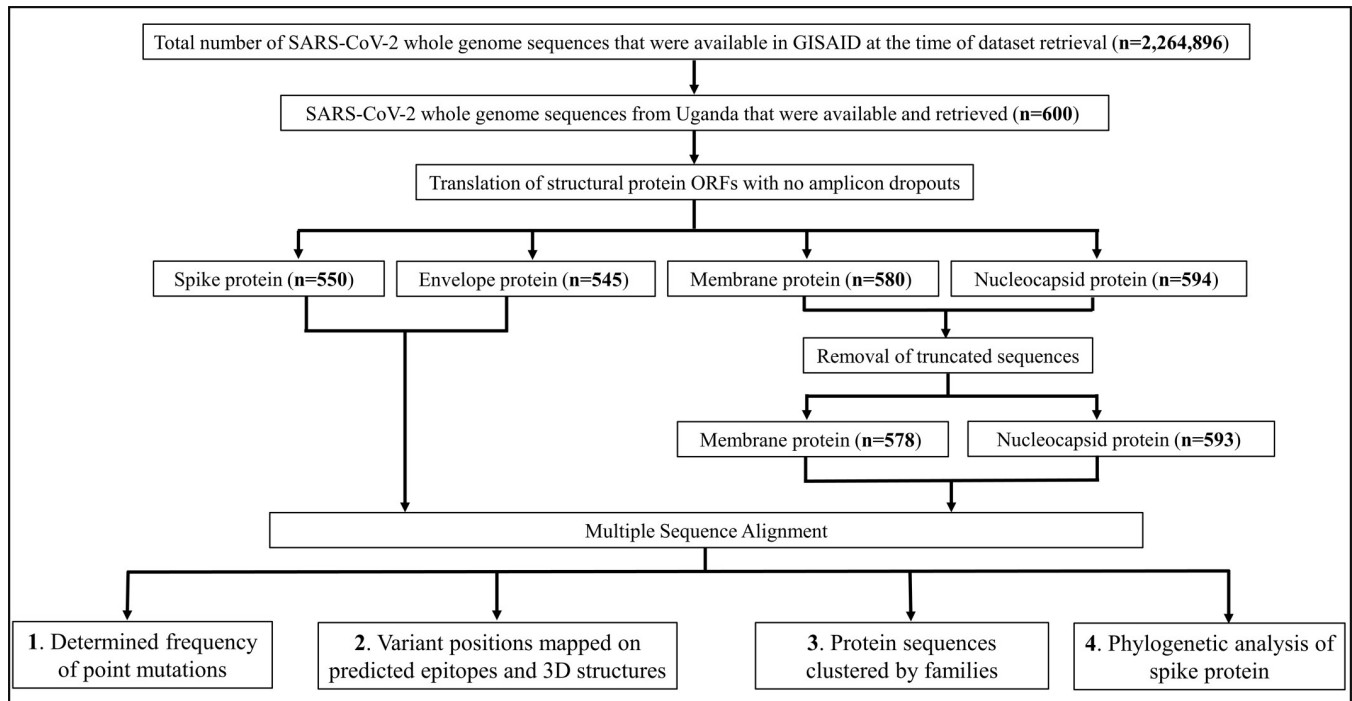

**Fig 1. SARS-CoV-2 structural protein heterogeneity analysis scheme.** SARS-CoV-2 whole genome sequences were retrieved from GISAID. ORFs were translated into structural protein followed by multiple sequence alignment. (**1**) Point mutations on positions on the alignments were analysed for residue composition and frequencies. (**2**) Variant positions were mapped on predicted epitopes and 3D structures. (**3**) Protein sequences sharing identity (100%) were clustered into families. (**4**) Phylogenetic analysis was conducted to probe the relationship between Ugandan spike protein sequences and Wuhan-Hu-1 wild-type (*wt*) or *Alpha*, *Beta*, *Delta* and *Gamma* the four spike variants of concern.

19 was detected in Uganda, to 24[th] June 2021. This dataset was accessed from the 23[rd] June to 9[th] August 2021. Data processing and analyses were sequentially performed as outlined below (**Fig 1**).

## Translation of open reading frames

SARS-CoV-2 genome sequences were translated into structural proteins using *getorf* software (EMBOSS, version 6.4.0.0) [35], and further validated with both *Geneious Prime*® software (Biomatters, version 2022.2.2) [36], and the National Center for Biotechnology Information (NCBI) ORF finder [37] with ORF set to a minimum length of 150 nucleotides. The tri-nucleotides ATG and UAG delineated the ORF start and stop codons, respectively. Those ORFs with amplicon drop-outs were removed from the analysis. Translated sequences were imported to *BioEdit Sequence Alignment Editor* software (Tom Hall, version 7.2.5) [38] where the sequences were inspected for defects including truncations.

## Multiple sequence alignment and analysis of variant positions

The *Geneious* Alignment package was used at the default setting (gap opening penalty = 12, gap extension penalty = 3, and refinement iterations = 2) during global alignment of translated sequences. Each of the alignment sets was exported to *BioEdit* where the extent of the variability of amino acid residue (entropy) at every position was calculated. The entropy values generated, per alignment set, by the *BioEdit* were plotted using *Microsoft Excel* version 2019.

Thereafter, frequencies of residues occupying each of the variant positions, per alignment set were plotted using *Microsoft Excel*.

## Mapping mutations within the epitopes and on the 3D structures

To have insight into the possible effects of the mutations encountered on performance of vaccines, immunodiagnostics and antibody-based therapies, mutated positions were mapped onto epitopes predicted in previous studies [39–42]. Also conducted was mapping of the mutant positions on 3D structures accessed from the PDB [43] and the NCBI [44] databases. *RasWin Molecular Graphics* software (GNU GPL, version 2.7.5.1) [45] was used for viewing the downloaded structures prior to cleaning water and heteroatoms using the *BOVIA Discovery Studio* client molecular software modelling (Dassault Systemes®, version 2021) [46]. The PDB IDs 7DDD, 6VYO, 6WJI and 7K3G were used for mapping point mutations on spike protein, nucleocapsid N-terminal domain (NTD), nucleocapsid C-terminal domain (CTD), and envelope protein, respectively. Although 3-D structure of membrane protein is not yet fully understood [47]; the structure was predicted from its primary sequence using the *Alpha-Fold2* protein 3D structure prediction software (Alphabet-DeepMind, version 2.1) [48] built in *UCSF ChimeraX* software (University of California, version 1.4) [49]. Afterwards, annotations of point mutations on the PDB structures were performed using *UCSF ChimeraX*.

## Classification of sequence families

The diversities of sequence families existing within each of the aligned protein sets were established. Alignments were imported to *BioEdit* and the sequences that were 100% identical were clustered into a family. Afterwards, the number of sequence families and the number of sequences per family (family size) were counted.

## Phylogenetic analysis of spike protein

The relationship between the Ugandan spike protein sequences and *wt*, or the *Alpha*, *Beta*, *Delta* and *Gamma* VOCs was investigated. A representative sequence was selected from each of the Ugandan spike sequence families to generate less crowded tree. To this sequence data, reference sequences i.e., Wuhan-Hu-1 *wt* (sp|P0DTC2|SPIKE SARS2), Alpha (QWE88920.1), Beta (QRN78347.1), Delta (QWK65230.1) and Gamma (QVE55289.1) sequences were added. The selected sequences were then analyzed using *Molecular Evolutionary Genetics Analysis* (MEGA) software (Pennsylvania State University, version 11) [50]. Briefly, the sequences were first aligned by *Multiple Sequence Comparison by Log-Expectation* (MUSCLE) software (drive5, version 3.8.31) using the default parameters. All the identified gaps were removed. The evolutionary history was then inferred by using the Neighbor-Joining and Maximum Likelihood methods with 1000 bootstrap. The evolutionary distances were computed using the Dayhoff matrix-based method [51].

## Ethics statement

This study was approved by the MAKCHS School of Biomedical Sciences Research Ethics Committee (*Approval number*: **SBS-2021-38**), and the Uganda National Council for Science and Technology (*Approval number*: **HS1706ES**). Given that secondary data was used, the need for consent was waived by the MAKCHS Sciences School of Biomedical Sciences Research Ethics Committee.

# Results

## Mutations on SARS-CoV-2 structural proteins

Six hundred genome sequences, which constituted 0.03% of the total SARS-CoV-2 genome sequences in GISAID nucleotide database (n = 2,264,896) as of 9th August 2021 were downloaded. Five Hundred and Seventy-Two (95.3%) of the sequences originated from MRC/UVRI & LSHTM, and 28 (4.7%) from the MAKCHS. Full-length ORFs on the genome coding for spike (n = 550), envelope (n = 545), membrane (n = 580), and nucleocapsid protein (n = 594) were translated into respective proteins. On inspection of the translated protein sets, N-terminally truncated membrane (n = 2) and nucleocapsid protein (n = 1) were discovered. The three defective sequences were dropped from the analysis reducing the number of membrane and nucleocapsid protein sequences to 578 and 593, respectively. Following alignment, positions harbouring variant amino acid residues were detected on spike (n = 137), envelope (n = 3), membrane (n = 9), and nucleocapsid protein sequences (n = 68). Entropy (H(x)) plot was generated to display positions harbouring variant residues and the degree of variability is shown as spiking bars (**Fig 2**). Spike protein sequence variant positions peak heights ranged from 0.01329 to 1.07852. The S1 subunit particularly the NTD, receptor binding domain (RBD) and the neighbouring region proximal to S1/S2 junction had higher density of tall peaks than the S2 subunit (**Fig 2A**). Specifically, conspicuous peaks were observed at positions

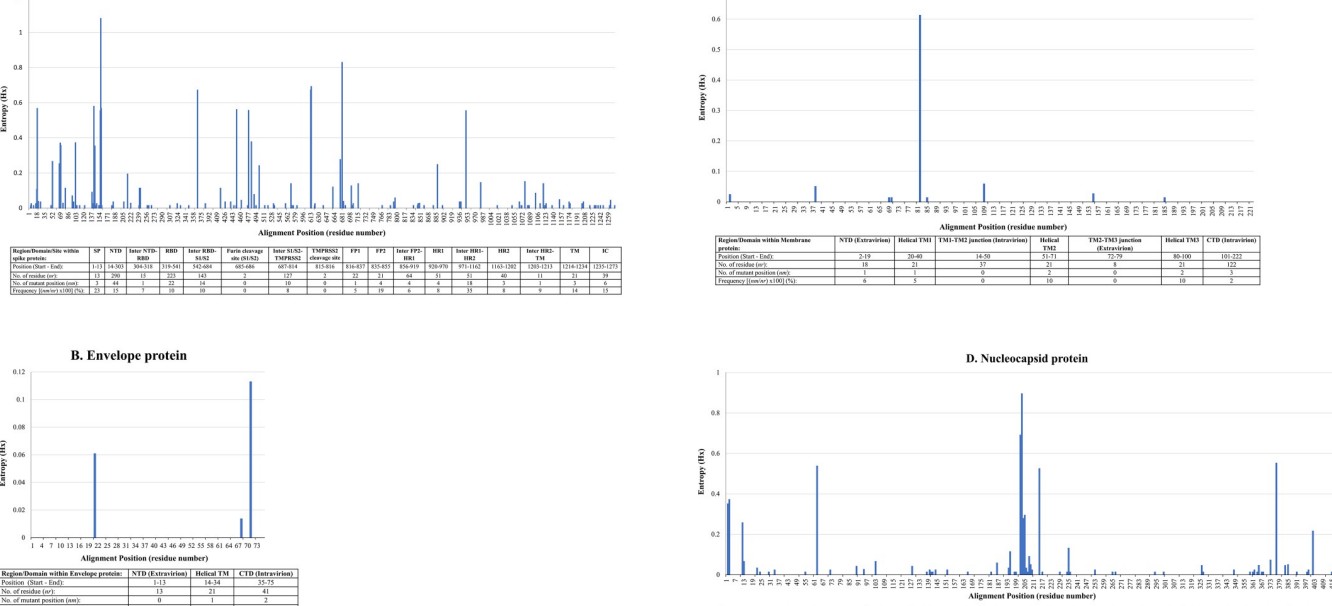

**Fig 2. Entropy (Hx) plot showing location of positions having residue mismatch on alignment of SARS-CoV-2 structural proteins and the distribution of variant positions over different domains/regions constituting the structural protein. (A)** Residue changes affected 137 of 1273 positions on spike protein. S1 subunit particularly the N-terminal domain (NTD), distal end of receptor binding domain (RBD), and positions proximal to the S1/S2 junction had higher concentration of tall peaks than the S2 subunit. **(B)** Envelope protein had only 3 of 75 positions affected by residue changes. **(C)** Multiple residue changes affected 9/222 positions across the membrane protein sequence. Except for position 82, residue changes on membrane proteins had low peak heights. **(D)** Residue changes on nucleocapsid sequence affected 68/419 positions. Whereas tall peaks were concentrated at the extremities (N and C-terminals) and centre of the nucleocapsid sequence, low peaks characterized the two RNA-binding domains. Table below each of the graphs shows the frequency of positions affected by point mutations along the entire length of the protein by region/domain/site. Definition of abbreviations used for salient domains/regions: SP, signal peptide; TMPRSS2, transmembrane serine protease 2; FP1, fusion peptide 1; FP2, fusion peptide 2; HR1, heptad repeat 1; HR2, heptad repeat 2; TM, transmembrane domain; IC, intracytoplasmic C-terminal domain of spike protein.

19 (T19I/R, entropy 0.56595), 142 (G142D, entropy 0.57827), 156 (E156G, entropy 0.55317), 157 (F157L/del, entropy 1.07852), 158 (R158S/del, entropy 0.56595), 367 (V367S, entropy 0.67074), 452 (L452R, entropy 0.55932), 478 (T478K, entropy 0.55524), 613 (Q613H, entropy 0.67074), 614 (D614G, entropy 0.69076), 681 (P681H/R, entropy 0.82789), and 950 (D950N, entropy 0.55317). Remarkably, in all the 550 spike protein sequences analysed both furin and TMPRSS2 cleavage sites were unaffected by point mutations. The envelope protein had short sparely distributed peaks confined to the transmembrane domain on position 21 (L21F, entropy 0.01339) and at the CTD on positions 68 (S68P, entropy 0,06059) and 71 (P71L, entropy 0.11268) (**Fig 2B**). The membrane protein had few randomly distributed variant positions with peak heights ranging from 0.01273 to 0.61174. The tallest peak was located at the transmembrane domain 3 (TM3) on position 82 (I82S, entropy value 0.61174). Interestingly, neither TM1-TM2 nor TM2-TM3 junction bore point mutation (**Fig 2C**). Peak heights on variant positions at the nucleocapsid protein sequence ranged from 0.01245 to 0.89423 (**Fig 2D**). Compared to the two RNA binding domains of the nucleocapsid protein sequence, tall entropy peaks were more concentrated at the N-arm, SR rich motif and C-arm. Specifically, these tall peaks were on positions 63 (D63G, entropy 0.53643), 202 (S202N, entropy 0.68944), 203 (R203K/M/N, entropy 0.89423), 215 (G215C, entropy 0.52377), and 377 (D377G/Y, entropy 0.55049).

## Composition and frequency of amino acid residues at variant positions

Amino acid changes at variant positions were either substitutions, deletions (Del) or both. A position *X* was said to be variant when at least one sequence from a collection of given structural protein sequences has a residue substitution or deletion. Except for positions 157, 614 and 681 on spike, *wt* residues predominated the variant positions of all the four structural proteins. For spike proteins, positions 157 (F157L/Del; 37.09% F, 38.73% L, and 24.18% Del), 614 (D614G; 46.55% D, and 53.46% G), 681 (P681H/R; 46% P, 3.82% H, and 50.18% R) had the least number of *wt* residue occupancy (<50%) followed by positions 367 (V367F; 60.55% V, and 39.45% F) and 613 (Q613H; 60.55% Q, and 39.45% H) at 60%, and then positions 19 (T19I/R; 75.6% T, 0.18% I, and 24.18% R), 142 (G142D/Del; 75.27% G, 24.36% D, and 0.36% del), 156 (E156G; 75.82% E, and 24.18% G), 158 (R158S/Del; 75.64% R, 0.18% S, and 24.18% del), 452 (L452R; 75.27% L, and 24.73% R), 478 (T478K; 75.64% T, and 24.73% K), and 950 (D950N; 75.82% D, and 24.18% N) at 75% (**Fig 3A**). None of the three variant positions on the envelope protein (L**21**F, S**68**P and P**71**L) had less than 90% *wt* residue occupancy, and position 82 (I82T; 69.9% I and 30.1% T) on membrane protein sequence had 70% *wt* residue occupancy (**Fig 3B and 3C**). On inspection of nucleocapsid protein sequences, a marked reduction in the number of *wt* residue was noticed at variant position 202 (S202N; 54.3% S and 45.7% N) followed by 203 (R203K/M/S; 64.25% R, 12.65% K, 22.93% M, and 0.17% S) and then positions 63 (D63G; 77.23% D, 22.77% G); 215 (G215C; 78.25% G, and 21.75% C), and 377 (D377G/Y; 76.9% D, 0.17% G, 22.93% Y). The rest of variant positions had ≥80% *wt* residue occupancy (**Fig 3D**).

## Mapping variant positions within the epitopes and spatial location on 3D structures

An assessment was conducted to predict the possible effects that these encountered point mutations may have on vaccines, immunoassays, and immunotherapies on the basis of their locations on the epitope sequences as well as 3D structures of each of the four structural proteins. Indeed, most mutated positions occurred within the predicted epitopes. We found that 91/137 mutated positions on spike protein were located within the epitopes, envelope had 3/3,

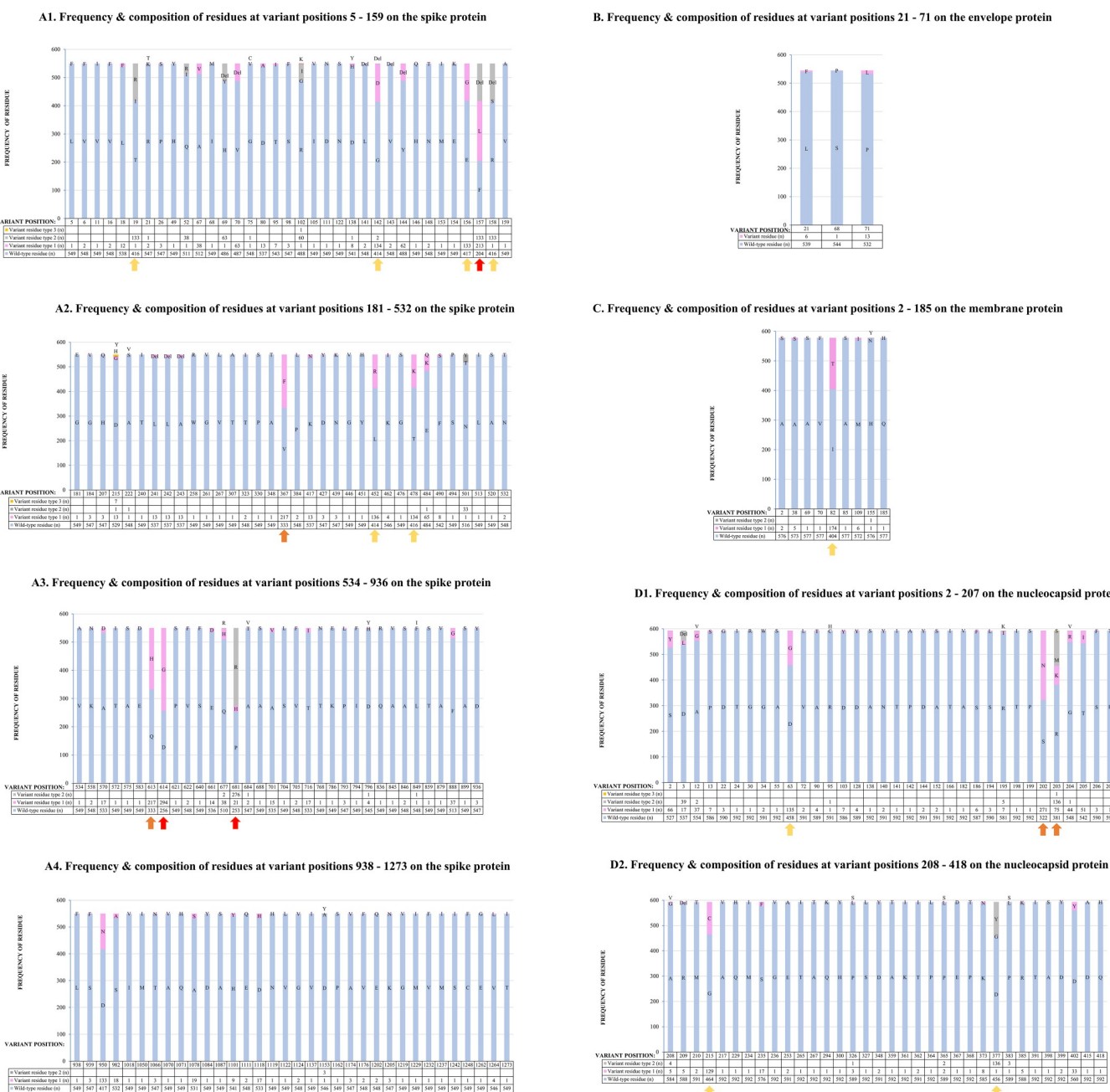

**Fig 3. A: Composition and frequency of residues, represented with one letter code, occupying each of the 137 variant positions on SARS-CoV-2 spike protein. A1**, composition and frequency of residues occupying variant positions 5–159; **A2**, composition and frequency of residues occupying variant positions 181–532; **A3**, composition and frequency of residue occupying variant positions 534–936; **A4**, composition and frequency of residues occupying variant positions 938–1273. Height of **light blue bars** qualifies the frequency of *wt*-like residues. Heights of **pink, grey and yellow bars** qualify the frequency of amino acid residue variants 1, 2 and 3, respectively. The a*rrows* are on positions where substantial reduction in the frequency of *wt*-like residues occurred. The *red*, *orange* and *yellow* arrows are on positions where frequency of *wt* residues is below 50%, at 60±5%, and at 70±5%, respectively. It is worthy noting that at any variant position there is Wuhan type residue and a substitution and/or Del being referred to as *wild-type residue* and *variant residue type*, respectively. As an example, at position 69 of spike protein sequences analyzed, there were 486 sequences bearing Wuhan type residue (*Wild-type residue*), one sequence had H69Y substitution point mutation (*Variant residue type 1*) and 63 sequences had H69Del (*Variant residue type 2*). Numerical assignment of variant residue types 1, 2 and 3 was random. **B and C: Composition and frequency of residues, represented with one letter code, occupying each of the three variant positions on the SARS-CoV-2 envelope protein (B) and nine variant positions on membrane protein (C)**. Height of the **light blue bars** qualifies the frequency of *wt*-like residues. The p**ink, and grey bars** qualify the frequency of variant residue type 1 and 2, respectively. **B**, three variant positions occurred on envelope protein on positions 21 (L21F), 68 (S68P) and 71 (P71L). **C**, nine variant positions were detected on the membrane protein. The *yellow arrow* on position 82 shows *wt*-like residue occupancy is within 70 ± 5%. At any variant position there is Wuhan type residue and a substitution being referred to as *wild-*

*type residue* and *variant residue type*, respectively. **D: Composition and frequency of amino acid residues, represented with one letter code, occupying each of the 68 variant positions on SARS-CoV-2 nucleocapsid protein. D1**, composition and frequency of residues occupying variant positions 2–207; and **D2**, compositions and frequency of residues occupying variant positions 208–418. Height of the **light blue bars** qualifies the frequency of *wt*-like residues. Heights of **pink, grey and yellow bars** qualify the frequency of variant residue type 1, 2 and 3, respectively. Five positions (63, 202, 203, 215 and 377) with low frequency of *wt*-like residues are shown. Positions 202 and 203 (*orange arrows)* had 54.3% and 64.2% *wt* residue, respectively. Positions 63, 215, and 377 (*yellow arrows*) had 77.2%, 78.2% and 76.9% *wt* residues, respectively. At any variant position there is Wuhan type residue and a substitution and/or Del being referred to *wild-type residue* and *variant residue type*, respectively.

membrane had 6/9, and the nucleocapsid had 46/68. Two spike variants, G446V and L452R, previously shown to resist convalescent sera and monoclonal antibody therapies, respectively [52] are surprisingly located within the predicted T-cell epitopes [40]. This information is summarized (**Table 1**).

Next, we mapped the variant sites on the 3D structures (**Fig 4**). PDB ID: 7DDD used for locating variant sites on the 3D of spike protein had 110/137 (80%) of variant positions. Whereas 55 of these variant sites were solvent exposed, 55 were located inside the protein core. The exposed variant sites were dispersed all-over the spike protein but the density was higher at the periphery of the dorsal surface (**Fig 4A**). Envelope protein had three variant sites; however, we were able to locate L21F only on the PDB ID: 7K3G luminal surface (**Fig 4B**). Due to the lack of available experimental structures, the structure of the membrane protein was predicted using *AlphaFold*. The structural model was perceived to be reliable based on the relatively high *AlphaFold* model confidence (pLDDT) scores (**S1 Fig**). All the nine variant positions could be located on the predicted structure. The S2F is located on the extravirion NTD; A69S, V70F, I82T and A85S are located on the transmembrane helix; and positions M109I, H155Y/N and Q185H are located on the intravirion CTD (**Fig 4C**). For the nucleocapsid protein, 36.8% (25/68) variants sites were located. Fourteen of these 25 variant sites were located on the NTD (PDB ID: 6VYO), and 11 were located on the CTD (PDB: 6WJI) (**Fig 4D**). While 11 out of the 14 variant sites located on the NTD were surface exposed, 10 of the 11 variant sites on CTD were surface exposed.

## SARS-CoV-2 structural protein sequence variants circulating in Uganda

Structural protein sequences that were 100% identical with respect to residues occupying a given position were clustered into a family. Each of the four structural proteins had multiple clusters depicting sequence heterogeneity. The spike protein formed the most numerous clusters (n = 141) followed by nucleocapsid (n = 81), membrane (n = 11) and envelope protein (n = 4). Further analysis was conducted to establish the size of each of the sequence families. For the spike protein, family 109 comprising typical *Delta* VOC was the largest and accounted for 16.5% (91/550) of the sequences followed by family 17 characterized by F157L, V367F and Q613H combined mutation (10.5%, 58/550); family 27 characterized by F157L, V367F, Q613H and P681R combined mutation (9.3%, 51/550); family 2 characterized by D614G single mutation (7.6%, 42/550); family 28 characterized by R102, F157L, V367F, Q613H and P681R combined mutation (7.1%, 39/550); family 1 comprising typical Wuhan-Hu-1 *wt* (4.4%, 24/550); family 79 characterized by Q52R, A67V, HV69-70 Del, Y144 Del, E484K, D614G, V551H and F888L combined mutation (4%, 22/550); and family 126 characterized by T19R, G142D, E156G, FR157-158 Del, L452R, T478K, D614G, P681R, D950N and A1078 combined mutation (2.5%, 14/550). The rest of the sequence families (n = 133) had fewer sequences ranging from 8 to 1 (1.5% to 0.2%). Among the notably small-sized spike sequence families were family 78 comprising typical *Beta* VOC (1.5%, 8/550) and family 96 comprising typical *Alpha* (1.1%, 6/550) (**S1 Table**). For the envelope protein, family 1 comprising only Wuhan-Hu-1 *wt* sequences was the most abundant (96.3%, 525/545) followed by family 4 characterized by

**Table 1. Point mutations located within the predicted epitopes on SARS-CoV-2 Wuhan-Hu-1 *wt* structural proteins.**

| Protein | Mutation(s) encountered on predicted epitopes | Sequence of the epitope affected | Start-End | Epitope type | Location on structural protein domain/region | Reference |
|---|---|---|---|---|---|---|
| Spike | L5F, V6F, V11I | FVF**L**VLLPL**V** | 2–11 | T cell (MHC-I) | S1 | [39] |
| | H49Y, Q52I/R | **H**ST**Q**DLFLPF | 49–58 | T cell (MHC-I) | S1 | [39] |
| | R102G/I/K, I105V, D111N | I**R**GWIFGTTL**D**SKTQSLL | 101–118 | T cell (MHC-II) | S1 | [40] |
| | N122S | QSLLIVN**N**ATNVVIK | 115–129 | T cell (MHC-II) | S1 | [39] |
| | D138H/Y, L141Del, G142D/Del, **V143Del**, **Y144Del**, H146Q, N148T, M153I, E154K, E156G, **F157L/Del**, R158S/Del, V159A | KVCEFQFCND**P**F**L**G**V**_**Y**_**Y**H**K**N**N**KSW**M**ESE**F**_**R**VY | 129–160 | B & T cell | S1 | [42] |
| | V143Del, Y144Del, H146Q, N148T | **V**YY**H**K**N**NKS | 143–152 | B cell | S1 | [39] |
| | G181E, G184V | QPFLMDLE**G**KQ**G**N | 173–185 | T cell (MHC-II) | S1 | [40] |
| | T240I, L241Del, L242Del, A243Del, W258R | TRFQ**T**LL**A**LHRSYLTPGDSSS**G**W | 236–258 | T cell (MHC-II) | S1 | [40] |
| | W258R, G261V, V267L | GDSSSG**W**TA**G**AAAYY**V**GYLQPRTFLLKYNENGT | 252–284 | B & T cell | S1 | [42] |
| | T307A | DAVDCALDPLSETKCTLKSF**T**VEKGIYQTSN | 287–317 | B cell | S1 | [40] |
| | | KSF**T**VEKGIYQTSNFRVQ | 304–321 | T cell (MHC-II) | S1 | [40] |
| | P330S | VRF**P**NITNLCPFGEVFN | 327–343 | B & T cell | S1 | [41] |
| | P384L | SASFSTFKCYGVSPTK**L** | 371–387 | T cell (MHC-II) | S1 | [40] |
| | **K417N** | RQIAPGQTG**K**_ | 408–417 | T cell (MHC-I) | S1 | [39] |
| | D427Y | KLP**D**DFTGCV | 424–433 | T cell (MHCI) | S1 | [40] |
| | G446V[a], Y451H, **L452R**[b] | NLDSKV**G**GNYNY**L**_YRLFR | 440–457 | T cell (MHC-II) | S1 | [40] |
| | Y451H, L452R, K462I | **Y**LYRLFRKSNL**K**PFERDI | 451–468 | T cell (MHC-II) | S1 | [40] |
| | K462I | **K**PFERDISTEIYQ | 462–474 | T cell (MHC-II) | S1 | [40] |
| | K462I, G476S, T478K, **E484K/Q**, F490S, S494P | **K**PFERDISTEIYQA**G**S**T**PCNGV**E**_GFNCY**F**PLQ**S** | 462–494 | B & T cell | S1 | [42] |
| | F490S, S494P | CY**F**PLQ**S**YGF | 488–497 | T cell (MHC-I) | S1 | [39] |
| | A520S | FELLH**A**PATV | 515–524 | T cell (MHC-I) | S1 | [39] |
| | N532T, V534A, K558N, A570D, T572I, A575S, E583D | VCGPKKST**N**L**V**KNKCVNFNFNGLTGTGVLTESNK**K**FLPFQQFGRDI**A**D**T**T**D**AVRDPQTL**E**ILDITPCSFGGVSVI | 524–598 | B cell | S1 | [40] |
| | Q613H, **D614G**, P621S, V622F, S640F | GTNTSNQVAVLY**Q**D**_VNCTEV**P**V**A**IHADQLTPTWRVYSTG**S** | 601–640 | B cell | S1 | [40] |
| | Q677H/R | DIPIGAGICASYQT**Q**TNS | 663–680 | B & T cell | S1 | [41] |
| | P681H/R, A684T/V, A688S | S**P**RR**A**RS**V**AS | 680–689 | T cell (MHC-I) | Overlaps S1 & S2 | [39] |
| | A701V, S704L, V705F | QSIIAYTMSLG**A**ENS**V**AY | 690–707 | T cell (MHC-II) | S2 | [40] |
| | | SLGA**E**NS**V**AY | 698–707 | T cell (MHC-I) | S2 | [39] |
| | K786E, P793L, I794F, D796H/Y | V**K**QIYKTPP**I**K**D**FGGFNF | 785–802 | T cell (MHC-II) | S2 | [40] |
| | F888G, A899S | **F**GAGAALQIPF**A**MQMAYRFNGI | 888–909 | B cell | S2 | [40] |
| | A899S | GAALQIPF**A**MQMAYRFN | 891–907 | B & T cell | S2 | [41] |
| | | P**F**AMQMAYRF | 897–906 | T cell (MHC-I) | S2 | [39] |
| | | PF**A**MQMAYRFNGIGVTQ | 897–913 | B & T cell | S2 | [41] |
| | D936V, L938F, S939F, D950N | **D**SL**S**STASALGKLQ**D**VV | 936–952 | T cell (MHC-II) | S2 | [40] |
| | S982A | VLNDIL**S**RL | 976–984 | T cell (MHC-I) | S2 | [40] |
| | I1018V | QLIRAAE**I**RASANLAATK | 1011–1028 | T cell (MHC-I) | S2 | [40] |
| | T1066N | VVFLHV**T**YV | 1060–1068 | B & T cell | S2 | [41] |
| | T1066N, A1070V, Q1071H | HV**T**YVP**A**QEK | 1064–1073 | T cell (MHC-I) | S2 | [39] |
| | H1101Y, E1111Q | **H**WFVTQRNFY**E**PQII | 1101–1115 | T cell (MHC-I) | S2 | [40] |
| | P1162S | KNHTS**P**DVDLGDISGIN | 1157–1173 | B & T cell | S2 | [41] |
| | A1174V, V1176F | DLGDISGIN**A**S**V**VNIQK | 1165–1181 | B & T cell | S2 | [41] |
| | E1202Q, K1205N | EIDRLNEVAKNLESLIDLQ**E**L**G**KYEQY | 1182–1209 | B & T cell | S2 | [41] |
| | G1219V | KWPWYIWL**G**F | 1211–1220 | T cell (MHC-I) | S2 | [39] |
| | E1262G, V1264L, T1273I | CKFDEDDS**E**PV**L**KGVKLHY**T** | 1254–1273 | B & T cell | S2 | [41] |

*(Continued)*

**Table 1.** (Continued)

| Protein | Mutation (s) encountered on predicted epitopes | Sequence of the epitope affected | Start-End | Epitope type | Location on structural protein domain/ region | Reference |
|---|---|---|---|---|---|---|
| Envelope | L21F | FL_AFVVFLLV | 20–29 | T cell (MHC-I) | Helical TM | [39] |
| | S68P, P71L | SRVKNLNSS**R**V**P** | 60–71 | B cell | CTD | [39] |
| Membrane | A2S | M**A**DSNGTITVEELKKLLEQWNLVI | 1–24 | B cell | Overlaps NTD & Helical TM1 | [40] |
| | A69S, V70F | TLACFVLA**AV** | 61–70 | T cell (MHC-I) | Helical TM2 | [39] |
| | M109I | ASFRLFARTRS**M**WSF | 98–112 | T cell (MHC-II) | Overlaps Helical TM3 & CTD | [39] |
| | | FRLFARTRS**M** | 100–109 | T cell (MHC-I) | CTD | [39] |
| | | FARTRS**M**WSF | 103–112 | T cell (MHC-I) | CTD | [39] |
| | H155N/Y | HLRIAGH**H**L | 148–156 | T cell (MHC-I) | CTD | [40] |
| | Q185H | TSRTLSYYKLGAS**Q**RV | 172–187 | B & T cell | CTD | [41] |
| Nucleocapsid | D3L/Del, A12G/V, P13S | **D**NGPQNQRN**AP** | 3–13 | B cell | N-arm | [39] |
| | A55S | PQGLPNNTASWFT**A**LTQHGKE | 42–62 | B cell | Overlaps N-arm & RNA binding domain | [40] |
| | V72L | FPRGQG**V**PIN | 66–75 | T cell (MHC-I) | RNA binding domain | [39] |
| | A90T | IGYYRR**A**TRRIRGGD | 84–98 | T cell (MHC-II) | RNA binding domain | [39] |
| | D103Y | GGDGKMK**D** | 96–103 | B cell | RNA binding domain | [39] |
| | A138S, N140Y, T141I, P142A, D144Y | **A**L**NTP**K**D**HI | 138–146 | T cell (MHC-I) | RNA binding domain | [40] |
| | T166I | NNNAATVLQLPQG**T**TLPKGF | 153–172 | B cell | RNA binding domain | [40] |
| | | LQLPQG**T**TL | 159–167 | T cell (MHC-I) | RNA binding domain | [40] |
| | A182V, S186F, S194L, R195I/K | AEGSRGGSQ**A**SS**R**S**S**SRNSS**R**NS | 173–197 | B cell | Overlaps RNA binding domain & SR-rich motif | [39] |
| | A182V, S186F | SRGGSQ**A**SS**R**S**S**SRSR | 176–191 | B & T cell | SR-rich motif | [41] |
| | G215C | AGNG**G**D | 211–216 | B cell | Inter SR-rich motif–Dimerization domain RNA binding | [39] |
| | G215C, A217V | **G**D**A**ALALLLL | 215–224 | T cell (MHC-I) | Inter SR-rich motif–Dimerization domain RNA binding | [40] |
| | Q229H | LLLDRLN**Q**L | 222–230 | T cell (MHC-I) | Inter SR-rich motif–Dimerization domain RNA binding | [40] |
| | Q229H, M234I | RLN**Q**LESK**M** | 226–234 | T cell (MHC-I) | Inter SR-rich motif–Dimerization domain RNA binding | [40] |
| | M234I, S235F, G236V | ESK**MSG**KGQQQQGQT | 231–245 | B cell | Inter SR-rich motif–Dimerization domain RNA binding | [39] |
| | E253A | QQQGQTVTKKSAA**E**ASKK | 240–257 | B & T cell | Overlaps Inter SR-rich motif–Dimerization domain RNA binding & Dimerization domain RNA binding | [41] |
| | T265I, A267T | A**TK**A**Y**NVTQAFGRRG | 264–278 | T cell (MHC-II) | Dimerization domain RNA binding | [39] |
| | | **TK**A**Y**NVTQAF | 265–274 | T cell (MHC-I) | Dimerization domain RNA binding | [40] |
| | Q294K, H300Y | IR**Q**GTDYK**H**WPQIAQFA | 292–308 | B & T cell | Dimerization domain RNA binding | [41] |
| | P326L/S, S327L | AQFAPSASAFFGMSRIGMEVT**PS**GTWLTYTGAI | 305–337 | T cell (MHC-II & I) | Dimerization domain RNA binding | [42] |
| | | FFGMSRIGMEVT**PS**GTW | 314–330 | B & T cell | Dimerization domain RNA binding | [41] |
| | | MEVT**PS**GTWL | 322–331 | T cell (MHC-I) | Dimerization domain RNA binding | [39, 40] |
| | D348Y | NFK**D**QVILL | 345–353 | T cell (MHC-I) | Dimerization domain RNA binding | [40] |
| | A359T, K361I, T362I, P364L, P365L/S, E367D, P368T, K373N, D377G/Y, P383L/S, R385K, T391I, A398S, D399Y | KHID**A**Y**KT**F**PP**T**EP**KKDK**K**KKT**D**EAQPL**PQR**QKKQP**T**VTLLPA**AD**LD | 355–401 | B cell | Overlaps Dimerization domain RNA binding & C-tail | [40] |
| | K361I, T362I, P364L, P365L/S, E367D, P368T, K373N | Y**KT**F**PP**T**EP**KKDK**K**KK | 360–375 | B & T cell | Overlaps Dimerization domain RNA binding & C-tail | [41] |
| | K361I, T362I, P364L, P365L/S, E367D, P368T | **KT**F**PP**T**EP**KK | 361–370 | T cell (MHC-I) | Overlaps Dimerization domain RNA binding & C-tail | [39] |
| | D415A | QSMSSA**D**S | 408–416 | B cell | Overlaps Dimerization domain RNA binding & C-tail | [39] |

The underlined amino acid residues located on epitope sequences have either undergone substitution or deletion. Potential influential positions annotated on the 3D structures are coloured.

[a]Mutation (G446V) shown to resist convalescent serum [52]

[b] Mutation (L452) shown to resist neutralizing monoclonal antibody [52].

**A. Spike protein homotrimer**: Chain **A,** white**;** Chain **B,** blue **&** Chain **C,** cyan

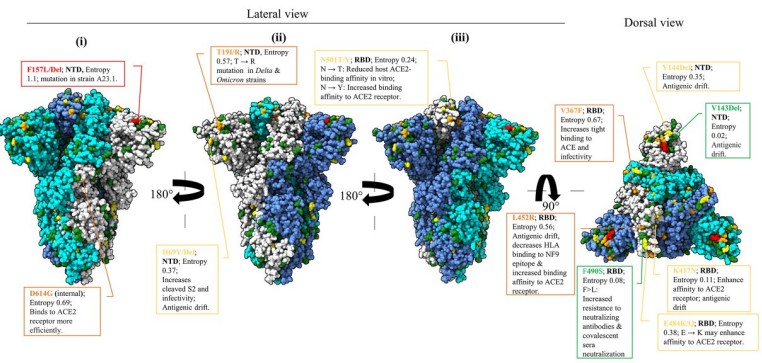

**B. Envelope protein pentamer**: Chain **A,** white; Chain **B,** blue; Chain **C,** cyan; Chain **D,** grey; & Chain **E,** orange

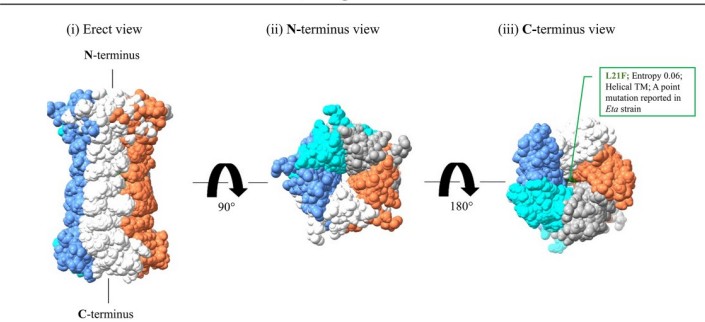

**C. Membrane protein monomer**: Chain **A** only (modelled by *AlphaFold*)

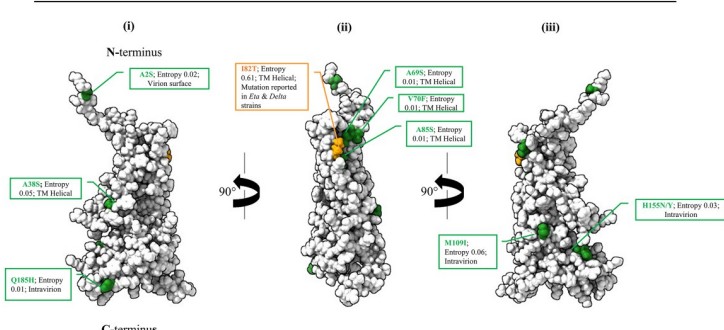

**D. Nucleocapsid protein multimer (NTD & CTD)**

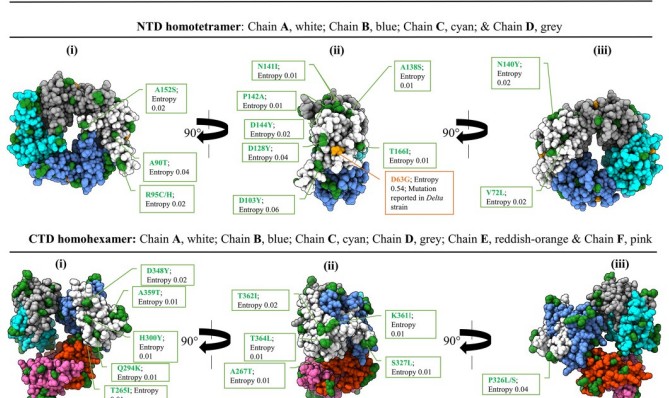

**Fig 4. Exposed variant positions on the 3D structure (chain A, coloured white) of SARS-CoV-2 structural proteins. (A) Spike protein** PDB ID: 7DDD, **(B) Envelope protein** PDB ID: 7K3G, **(C) Membrane protein (Chain A) structure generated by** *AlphaFold2*, **and (D) Nucleocapsid protein N-terminal domain** PDB ID: 6VYO **or C-terminal domain** PDB ID: 6WJI. Coloured amino acid residues on Chain A indicate positions with entropy value above zero. **Red**, entropy value ≥ 1; **orange**, entropy value ≥ 0.5 ≤ 1; **yellow** entropy value <0.5 ≥0.1; and **green** entropy value <0.1. (**A**) variant positions were dispersed all-over spike protein surface *albeit* positions with high entropies being more concentrated on the dorsum. (**B**) variant position 21 (L21F) on envelope protein was mapped to the luminal surface of the pentameric ion channel. (C) Except for a single variant position located on the extravirion of the membrane protein, the rest were either at the transmembrane junction or intravirion. (**D**) variant positions were evenly distributed on the surface of both N- and C-terminal domains of nucleocapsid. However, N-terminal domain had variant positions with relatively high entropy values (yellow) unlike the C-terminal where all the variant positions had low entropy values below 0.1 (green).

P71L single mutation (2.4%, 13/545), family 2 characterized by L21F single mutation (1.1%, 6/545), and the least being family 3 characterized by S68P single mutation (0.2%, 1/545) (**S2 Table**). Likewise, sequence family 1 in the membrane protein group comprising typical Wuhan-Hu-1 *wt* had the largest size (68.4%, 397/580) followed by family 9 characterized by I82T single mutation (28.4%, 165/580); family 5 characterized by I82T and M109I combined mutation (1%, 6/580); and family 8 characterized by A38S single mutation (0.9%, 5/580). The rest of families comprised of single member each contributing 0.2% of the entire membrane protein sequence collection. We characterized family 5 by V70F and H155Y combined mutation, family 3 by A2S and I82T combined mutation, family 4 by I82T and H155N combined mutation, family 6 by I82T and Q185H combined mutation, family 7 by A2S single mutation, family 10 by A69S single mutation and family 11 by A85S single mutation (**S3 Table**). Of the nucleocapsid protein sequences, family 2 characterized by S202N single point mutation, had the largest size (34.5%, 205/594) followed by family 71 characterized by D63G, R203M, G215C, D377Y combined mutation (18.9%, 112/594), family 1 comprising typical Wuhan-Hu-1 *wt* (5.9%, 35/594), family 52 characterized by S2Y, D3Del, A12G, T205I combined mutation (5.2%, 31/594), family 11 by S2Y, S202N, R203K combined mutation (4.2%, 25/594), and family 9 by D402Y single point mutation (3.9%, 23/594). The rest of the families (n = 75) had between 8 and 1 members (**S4 Table**).

## Phylogeny of the spike protein

The relationship between the Uganda SARS-CoV-2 spike protein sequences (n = 550) and Wuhan-Hu-1 *wt* (P0DTC2) reference strain or VOCs including *Alpha* (QWE88920.1), *Beta* (QRN78347.1), *Delta* (QWK65230.1) and *Gamma* (QVE55289.1) was assessed in order to establish which of the strain (s) was circulating in Uganda. Uganda sequences clustered with Wuhan-Hu-1 *wt*, the *Alpha*, *Beta*, and *Delta* VOCs but not *Gamma* (**S2 Fig**). Two hundred and fifty-six sequences (from 62 families) clustered with Wuhan-Hu-1 *wt*, 133 sequences (from 20 families) clustered with *Delta* VOC, 17 sequences (from 10 families) clustered with *Alpha* VOC (*orange-brown* dot), and 13 sequences (from six families) clustered with *Beta* VOC. Then there were 131 sequences (from 43 families), which neither clustered with Wuhan-Hu-1 *wt* nor the 3 spike VOCs (*Alpha*, *Beta*, and *Delta*). Of these un-clustered sequences, 14 (from 7 families) strongly clustered with family 79, which comprised of 22 sequences.

## Discussion

The genome of SARS-CoV-2 virus has accumulated several mutations [53], which have decreased the performance of diagnostics and therapeutic antibodies. For these reasons, refinement of first-generation COVID 19 medical products in tandem with emerging virus variants

is required. Using SARS-CoV-2 structural protein sequences originating from Uganda as a case study, retrospective profiling was conducted to ascertain degree of heterogeneity, which occurred between March 2020 to June 2021.

Although mutations affected multiple positions on each of the structural proteins, spike and nucleocapsid proteins were the most affected. The S1 subunit (mostly NTD, RBD and the region upstream the S1/S2 junction) of spike protein was more affected than the S1 in agreement with Jia and Gong [54]. The volatility of residues located on a more exposed S1 subunit allows the virus to thwart antibody neutralization [55] thereby promoting transmission [56]. Unlike S1, the concealed S2 subunit [57] was more conserved. Owing to its crucial roles in stabilization of the spike protein architecture [57] and host cell membrane fusion [13, 58], extensive mutation of S2 could be detrimental to the virus. Similarly, high fidelity of both furin and TMPRSS2 cleavage sites explains their crucial role in proteolytic activation of SARS-CoV-2 virus for host cell entry and mutation of any of these two sites is lethal [16]. Hence, the invariant S2 subunit offers opportunity for development of cross-reactive protein-based vaccines, immunoassays and immunotherapies compared to S1. However, inaccessibility of S2 by large therapeutic molecules may present a problem. Low molecular weight therapeutic compounds such as nanobodies [59] and antimicrobial peptides [60] overcome obstructed access to such buried targets on the virus. Unlike spike or nucleocapsid, envelope and membrane proteins were more conserved throughout their entire lengths. Conservation of both envelope and membrane proteins is in order given their concealment from neutralizing antibodies. Moreover, the lengths of these two proteins are relatively short [47] meaning their mutations can result in functional impairment leading to the loss of viral fitness as shown by Verdiá-Báguena et al. [61] that mutations N15A and V25F impair ion conductivity of the envelope protein. The invariant nature of envelope and membrane proteins offer suitable targets where cross-reactivity is required. For nucleocapsid protein, the RNA binding domains were conserved although their flanking regions were variant as it was previously reported [62]. Relative conservation of RNA binding domains is attributable to strict selection of residues for specific RNA interacting residues, which is not the case with residues located in the variant regions. A consequence of mutation on nucleocapsid is antigenic drift, which has led to false-negative test result by nucleocapsid-based commercial tests [27, 63]. Assessment of the performance of panels of nucleocapsid-based reagents on recombinant forms of predominant variants documented in this study is therefore highly recommended.

Outstanding variant positions on the spike protein were F157L/Del at the NTD, D614G located distal to the RBD, and P681H/R located proximal to the S1/S2 furin cleavage site. F157L/Del variant characterizes SARS-CoV-2 virus lineage A.23.1 detected in Uganda [64]. It is presumptive to link the dominance of F157L/Del to immune escape given its location on a predicted epitope published elsewhere [42]. On the other hand, co-evolution of F157L/Del with P681H/R variant, which is known for promoting cell membrane fusion [65], could have enhanced infectivity resulting in proliferation of the variant. D614G global dominance was reported earlier [66, 67] and it is associated with increased infectivity [66] ascribed to re-configured RBD, which favors ACE2 receptor binding [67]. Given that existing vaccines, diagnostics and neutralizing antibodies panels were raised against Wuhan-Hu-1 *wt* targets, extensive validation of these products on F157L/Del, D614G and P681H/R variants is, therefore, highly recommended. Although they have not surpassed *wt*, V367F and Q613H spike variants require follow-up because of their apparent rising levels. Apparent increase in the frequency of V367F spike variant contradicts reports that it is sensitive to neutralizing antibodies [52]. Low herd immunity at the start of the pandemic in combination with co-evolution of V367F with fusion promoting P681H/R variants could explain observed sharp rise. A sharp rise in Q613H variant is speculated to be associated with increased transmissibility following re-configuration of

RBD like the D614G, and co-evolution with the P681H/R variant, which promotes cell fusion. Apart from dominant spike variants, underrepresented variants encountered require close monitoring to avert a possible buildup into a next pandemic. For example, Li et al. [52] has shown that low frequency L452R variant located on predicted T-cell epitopes [40] resists antibody neutralization showing the variant harbors capability of proliferation to epidemic levels. Envelope protein variants (L21F, S68P or P71L) were remarkably lower than the *wt*. P71L variant, which was the most predominant among the envelope protein variants co-evolves with the *Beta* spike VOC [68], and L21F variant, the second most predominant variant, co-evolves with the *Eta* spike variant of interest [69]. Thus, association with highly transmissible spike variants explain relative high proliferation of P71L and L21F envelope sequence variants. Membrane protein had nine variant positions and I82T being the most highly represented (**Fig 3C**). Co-evolution of I82T sequence variant with the highly transmissible *Delta* spike variant [70] explains its high frequency. Worth noting, frequencies of other membrane protein variants were extremely low irrespective of their topology (**Figs 2C and 4C**) and location within epitopes (**Table 1**) meaning that these two attributes may not have much influence on their propagation. The most predominant variant S202N on nucleocapsid protein co-evolves with a highly transmissible lineage A.23.1 spike variant [64], and the second most predominant variant R203K/M co-evolves with the *Theta*, *Omicron* and *Delta* spike variants [71]. High proliferation of S202N and R203K/M, both co-evolving with high transmissible spike variants, shows the positive influence of spike protein has on propagation of other SARS-CoV-2 structural proteins. Collectively, it is now apparent that mutation of spike protein to a highly transmissible variant drives amplification of remotely located co-evolving variants.

Protein sequences were grouped by family based on 100% identity of residues at each of the positions. Spike protein formed the most diverse clusters totaling to 141 families followed by nucleocapasid (n = 81), membrane (n = 11) and envelope (n = 4). Of the spike protein sequence families, typical Wuhan-Hu-1 *wt* family had unexpectedly few members (n = 24) representing 4.36% (24/550) of the entire spike sequences recorded. Wuhan-Hu-1 *wt* family was present at the beginning of the pandemic and quiesce by October, 2020. Low frequency of Wuhan-Hu-1 *wt* spike sequence is mostly attributed to transmission interruption due to strict implementation of public health measures at the very beginning of the pandemic. While the *wt* strain wanes, new spike variants emerged and became dominant causing mild to severe disease. The new variants were able to spread rapidly for two major reasons: (1) laxity in the implementation of public health measures, and (2) resistance to herd immunity induced by natural exposure as well as vaccine. Thus, the latter can be overcome through accelerating vaccine coverage employing next-generation spike variant derived cocktail vaccine. Majority of envelope protein (96.3%) and membrane protein (68.4%) sequences were *wt*. Next to Wuhan *wt*, the outstanding membrane protein sequence family was I82T variant. I82T sequence variant co-evolves with the highly transmissible *Delta* spike variant explaining its high prevalence in the population. Like spike, typical Wuhan-Hu-1 nucleocapsid protein sequence was poorly represented accounting for 5.9% (35/593) of the entire sequences. This typical Wuhan-Hu-1 nucleocapsid protein sequence could not be detected by September 2020 coinciding with the disappearance of Wuhan-Hu-1 *wt* spike protein sequence signifying that SARS-CoV-2 viruses possessing parent spike as well as nucleocapsid proteins may have loss fitness in the course of the pandemic. Nucleocapsid sequence variant S202N has the most predominant sequence family with 205 members (34.5%). As it was noted earlier, S202N sequence variant is highly amplified courtesy of co-evolution with a highly transmissible lineage A.23.1 spike variants. Collectively, observed rapid evolution particularly of the spike and nucleocapsid sequences calls for rapid refinement of Wuhan-Hu-1 *wt* based vaccines, diagnostics and immunotherapy to incorporate predominant and fixated sequence families to catch up with the pace of virus

evolution. Where target conservation is required for cross-reactivity, envelope and membrane protein are suitable candidates. However, the use of vaccines, diagnostics and therapeutics designed based on *wt* sequence information should not yet be discouraged without gathering concrete proofs through repeated experimental evidence.

The evolutionary relationship between circulating spike protein sequences and Wuhan-Hu-1 *wt* or the *Alpha*, *Beta*, *Gamma* and *Delta* VOCs was assessed. The majority of sequences clustered with Wuhan-Hu-1 spike followed by *Delta*, *Alpha*, and *Beta*. *Gamma* and related sequences were absent from the 550 sequences examined. There were other large groups of sequences that neither clustered with Wuhan-Hu-1 *wt* nor the VOCs. The observed sequence clustering patterns is not surprising. The majority of the sequences circulating in Uganda were closely related to Wuhan-Hu-1 *wt* given that sequence dataset where from samples collected from the first wave and immediately before the second wave when mutations were not yet extensive. Besides, at the beginning of the pandemic there was mandatory hospitalization and intensive case surveillance, which allowed collection of many Wuhan-Hu-1 *wt* related sequences. Also encountered in the dataset were the typical VOCs and related sequences, which emerged later in the course of the pandemic. These VOCs appeared in the trough lying between the crests of the first and second waves [72]. The *Alpha* and *Beta* clusters were much lower than *Delta*. *Alpha* and *Beta* VOCs appearance coincided with the time when Uganda was observing strict public health measures, which limited their transmission consequentially diminishing their population. Also, subclinical infections, which did not lead to hospitalization accounted for low recovery of SARS-CoV-2 virus variants causing mild infections. On the other hand, the *Delta* variants entered Uganda several months after the first lockdown when the biosecurity measures were no longer being maximally observed. This factor led to rapid transmission of SARS-CoV-2 variants, *Delta* variant inclusive, circulating at the time culminating in a second infection wave. Moreover, the virulent nature of the *Delta* spike variant led to massive hospitalization maximising the chances of sample collection for sequencing. It can be argued that public health measures instituted at the beginning of the pandemic followed by the implementation of vaccination programme greatly influenced the transmission of SARS-CoV-2 *wt* and other variants in Uganda. Therefore, the dynamics of SARS-CoV-2 variants described herein defines a Uganda situation, which may sharply vary from other countries.

## Conclusion

We showed that SARS-CoV-2 viruses that were circulating in Uganda within the study period had heterogenous structural proteins. Firstly, the findings of this surveillance study will contribute to the body of knowledge required for research and development of COVID-19 next-generation medical products targeting emerging SARS-CoV-strains. Secondly, the study provides baseline data for evaluating and measuring evolution of SARS-CoV-2 variants on a time scale. Thirdly, the investigation highlighted the dynamics of SARS-CoV-2 structural protein variants, which would guide policy makers on the choices of vaccines, test platforms and therapeutics befitting SARS-CoV-2 virus strains in circulation. The study was limited by the number of sequences analyzed, which were below the total number of COVID-19 reported cases in Uganda (n = 1,249) as of 9[th] August 2021 available at [72]. Firstly, it is recommended that global sequence dataset representing cases which occurred in the country be analyzed. Secondly, experimental data should be generated to foster the evidence-based understanding of the impact of encountered SARS-CoV-2 structural protein mutations on the course and control of COVID-19 disease. Thirdly, data from this study may not mirror and/or reflect the situation in other countries; therefore, it is recommendable that every country performs a comprehensive analysis of SARS-CoV-2 mutation trends.

## Supporting information

**S1 Fig. AlphaFold modelled structure of SARS-CoV-2 membrane protein.**
(TIF)

**S2 Fig. A phylogenetic tree showing the evolutionary relationships between SARS-CoV-2 spike protein sequences detected in Uganda and Wuhan-Hu-1 wt (sp|P0DTC2|SPIKE SARS2), or Alpha (QWE88920.1), Beta (QRN78347.1), Delta (QWK65230.1) and Gamma (QVE55289.1) variants of concern (VOCs).** Ugandan spike protein sequences clustered with Wuhan-Hu-1 *wt* (green dot) and three VOCs namely *Alpha* (orange-brown dot), *Beta* (blue dot), and *Delta* (red dot) but not *Gamma* (magenta dot). Numbers on the branches are bootstrap values. Crowding of the tree was avoided by showing only those bootstrap values >50. There were sequences, which neither clustered with Wuhan-Hu-1 reference strain nor the VOCs (n = 131). Some of these "un-clustered" sequences formed a separate cluster around family 79 (*yellow* dot). Thus, from largest to smallest cluster we had Wuhan cluster (n = 256), *Delta* VOC cluster (n = 133), *Alpha* VOC cluster (n = 17), and *Beta* VOC cluster (n = 13).
(PDF)

**S1 Table. Spike protein sequences clustered into families based on 100% amino acid residue identity.** Out of 141 families realized, *Delta* variant family had the largest size (n = 91).
(XLSX)

**S2 Table. Envelope protein sequences clustered into families based on 100% amino acid residue identity.** Out of four families realized, Wuhan-Hu-1 *wt* family had the largest size (n = 525).
(XLSX)

**S3 Table. Membrane protein sequences clustered into families based on 100% amino acid residue identity.** Out of 11 families realized, Wuhan-Hu-1 *wt* family had the largest size (n = 397).
(XLSX)

**S4 Table. Spike protein sequences clustered into families based on 100% amino acid residue identity.** Out of the 81 families realized, variant S202N family had the largest size (n = 205).
(XLSX)

## Acknowledgments

We acknowledge the contributions of the following personnel: Mr. Godwin Byamugisha Sabiiti, the project Monitoring and Evaluation Officer; Mr. Mordecai Tayebwa, the project Grants and Finance Manager; Mr. Anthony Nshimye, the Project Assistant coordinator. We also acknowledge the Government of Uganda for funding this study through the **Makerere University Research and Innovation Fund** (grant number **MAK-RIFDVCFA/026/20**). Finally, we acknowledge the MRC/UVRI/LSTM and the MAKCHS laboratories for making available in GISAID database the SARS-CoV-2 genome sequences from Uganda.

## Author Contributions

**Conceptualization:** Steven Odongo, Hedmon Okella, Moses Okee, Monica Namayanja, Brian Mujuni, Yann G. J. Sterckx, Dennison Kizito, Julius Julian Lutwama, Charles Ibingira.

**Data curation:** Steven Odongo, Hedmon Okella, Christian Ndekezi, Yann G. J. Sterckx.

**Formal analysis:** Steven Odongo, Hedmon Okella, Christian Ndekezi, Monica Namayanja, Yann G. J. Sterckx, Kokas Ikwap, Frank Nobert Mwiine.

**Funding acquisition:** Steven Odongo, Moses Okee, Brian Mujuni.

**Investigation:** Steven Odongo, Hedmon Okella, Christian Ndekezi, Monica Namayanja, Yann G. J. Sterckx, Dennison Kizito, Magdalena Radwanska, Stefan Magez, Kokas Ikwap, Frank Nobert Mwiine, Julius Julian Lutwama, Charles Ibingira.

**Methodology:** Steven Odongo, Hedmon Okella, Christian Ndekezi, Moses Okee, Monica Namayanja, Yann G. J. Sterckx, Dennison Kizito, Magdalena Radwanska, Stefan Magez, Kokas Ikwap, Frank Nobert Mwiine, Julius Julian Lutwama, Charles Ibingira.

**Project administration:** Steven Odongo, Moses Okee, Brian Mujuni, Dennison Kizito, Magdalena Radwanska, Julius Julian Lutwama, Charles Ibingira.

**Resources:** Steven Odongo, Moses Okee, Brian Mujuni, Magdalena Radwanska, Stefan Magez, Frank Nobert Mwiine, Julius Julian Lutwama, Charles Ibingira.

**Software:** Steven Odongo, Hedmon Okella, Christian Ndekezi, Yann G. J. Sterckx, Magdalena Radwanska, Stefan Magez, Kokas Ikwap.

**Supervision:** Steven Odongo, Moses Okee, Brian Mujuni, Frank Nobert Mwiine.

**Validation:** Steven Odongo, Hedmon Okella, Christian Ndekezi, Monica Namayanja, Brian Mujuni, Yann G. J. Sterckx, Magdalena Radwanska, Stefan Magez, Kokas Ikwap, Frank Nobert Mwiine, Charles Ibingira.

**Visualization:** Steven Odongo, Hedmon Okella, Christian Ndekezi, Yann G. J. Sterckx, Kokas Ikwap, Julius Julian Lutwama.

**Writing – original draft:** Steven Odongo, Hedmon Okella, Christian Ndekezi.

**Writing – review & editing:** Steven Odongo, Hedmon Okella, Christian Ndekezi, Moses Okee, Monica Namayanja, Brian Mujuni, Yann G. J. Sterckx, Dennison Kizito, Magdalena Radwanska, Stefan Magez, Kokas Ikwap, Frank Nobert Mwiine, Julius Julian Lutwama, Charles Ibingira.

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
