## [Decision Letter · Decision Letter 0]

18 Nov 2022

PONE-D-22-27027Retrospective in silico mutation profiling of SARS-CoV-2 structural proteins circulating in Uganda by July 2021: towards refinement of COVID-19 disease vaccines, diagnostics, and therapeuticsPLOS ONE

Dear Dr. Odongo,

Thank you for submitting your manuscript to PLOS ONE. After careful consideration, we feel that it has merit but does not fully meet PLOS ONE’s publication criteria as it currently stands. Therefore, we invite you to submit a revised version of the manuscript that addresses the points raised during the review process.

We look forward to receiving your revised manuscript.

Kind regards,

Maemu Petronella Gededzha, Ph.D

Academic Editor

PLOS ONE

Journal Requirements:

Additional Editor Comments:

Include the entropy value in the y axis for Figure 2A

Please list the sources and version of all software used in the methodology eg Muscle

The quality of the figures are poor

Revise phylogenetic tree (Figure 5): Exclude bootstrap value less than 70% and collapse some of the branches to improve the figure

The abstract need to be rewritten to also include the method used. The abstract is not attractive; it should arouse curiosity in the reader and should be clearer. It should not start with ‘Whereas”

Reviewers' comments:

Reviewer's Responses to Questions

**Comments to the Author**

1. Is the manuscript technically sound, and do the data support the conclusions?

Reviewer #1: Yes

Reviewer #2: Yes

2. Has the statistical analysis been performed appropriately and rigorously? 

Reviewer #1: Yes

Reviewer #2: Yes

3. Have the authors made all data underlying the findings in their manuscript fully available?

Reviewer #1: Yes

Reviewer #2: Yes

4. Is the manuscript presented in an intelligible fashion and written in standard English?

Reviewer #1: Yes

Reviewer #2: Yes

5. Review Comments to the Author

Reviewer #1: The protocols are adequately provided. The entire manuscript is well organized and clear enough to be accessible to non-specialists. There is enough information and the methods have been described sufficiently for other researchers to reproduce the experiments. The claims are properly placed in the context of the previous literature review. the content of the work flows and it is written in standard English.

An important fact has been raised by the authors in their discussion. That is recommendation of the extensive validation of the existing vaccines, diagnostics and neutralizing antibody panels on the three identified variants (F157L/Del, D614G and P681H/R) carried by the spike protein. This will assist in catching-up of the rapidly evolving virus.

Reference number 21 is omitted in the content.

Line 475 reference using the number like in content.

Line 486 spelling of beginning should be corrected.

Reviewer #2: The manuscript, describes of Retrospective in silico mutation profiling of SARS-CoV-2 2 structural proteins circulating in Uganda towards refinement of COVID-19 disease vaccines, 4 diagnostics, and therapeutics

The introduction is very comprehensive and well written. The authors were consistent and applied proof reading skills throughout the document. The methodology section answers the research very well and well detailed. The result were fully described and well detailed.

I find it very fair to congratulate the authors, it was very interesting to review this well-written paper

Minor comments needs to be dealt with

1. The author has reference 21 which was not cited anywhere in the article

2. Line 475 , the author used Vancouver referencing throughout the document, however, on this line they did not do so.

3. Line 486 beginning, there’s a typo error

Well done to the authors!!!

6. PLOS authors have the option to publish the peer review history of their article (what does this mean?). If published, this will include your full peer review and any attached files.

Reviewer #1: No

Reviewer #2: No

---

## [Author Response · Author response to Decision Letter 0]

3 Dec 2022

Response to the Academic Editor and the Reviewers:

RESPONSE TO ACADEMIC EDITOR

Comment #1: Please ensure that your manuscript meets PLOS ONE's style requirements, including those for file naming.

Response to comment#1: We have ensured strict adherence to all the PLOS ONE’s style requirements throughout the manuscript including file naming.

Comment#2: 2. Please provide additional details regarding participant consent. In the ethics statement in the Methods and online submission information, please ensure that you have specified what type you obtained (for instance, written or verbal, and if verbal, how it was documented and witnessed). If your study included minors, state whether you obtained consent from parents or guardians. If the need for consent was waived by the ethics committee, please include this information.

Response to comment#2: This was a “nested study” under “master study” titled “Development and validation of multiplex lateral flow assay for detection and differentiation of from SARS-CoV-2 and other human coronaviruses”, ref. no. SBS-2021-38. Because the entire study used secondary data (including the sequence dataset analyzed), we were granted consent waiver by the Makerere University College of Health Sciences (MAKCHS) School of Biomedical Sciences Research Ethics Committee. Please find accompany consent waiver attached. We have now modified the Ethics statement by incorporating additional statement on consent waiver. This statement has been included in the Method section of the Manuscript without track changes (line 211-216.), and it has been submitted online. 

Comment#3: Please review your reference list to ensure that it is complete and correct. If you have cited papers that have been retracted, please include the rationale for doing so in the manuscript text, or remove these references and replace them with relevant current references. Any changes to the reference list should be mentioned in the rebuttal letter that accompanies your revised manuscript. If you need to cite a retracted article, indicate the article’s retracted status in the References list and also include a citation and full reference for the retraction notice. 

Response to comment#3: We have thoroughly revised citations in the text against bibliography, therefore as it stands, there are no mismatches. 

Comment#4: Include the entropy value in the y axis for Figure 2A

Response to comment#4: Initially, we used graphs generated automatically by the BioEdit. However, BioEdit does not offer interactive interface, which does not allow manual manipulation by the end users. To include entropy values in the y-axis of Figure 2A, re-plotted all the graphs in Figure 2A-D graphs using the Microsoft Excel version 2019. 

Comment#5: Please list the sources and version of all software used in the methodology eg Muscle

Response to comment#5: Initially, the sources and/or versions of some software used in the study were not mentioned. We have rectified this omission by ensuring that all the software mentioned have sources and version numbers. 

Comment#6: The quality of the figures are poor

Response to comment#6: We strived to ensure that the current version of the figures will meet the taste of our readers. The quality of the figures was improved by increasing the dpi from the 96 to 600.

Comment#7: 7. Revise phylogenetic tree (Figure 5): Exclude bootstrap value less than 70% and collapse some of the branches to improve the figure

Response to comment#7: We acknowledge your concern on improving the clarity of the figure by collapsing some of the branches. However, collapsing the branches of the tree to some extent distorted the formation contained therein given that each of the OTUs is a separate entity representing 141 spike sequence variants encountered. Therefore, to avoid losing information, we decided to leave the figure intact; however, we excluded it for the body of the manuscript and instead included it as supporting information (S2 Fig.)

Comment#8: The abstract need to be rewritten to also include the method used. The abstract is not attractive; it should arouse curiosity in the reader and should be clearer. It should not start with ‘Whereas”

Response to comment#8: We highly appreciate this suggestion. We re-wrote the abstract to address this concern.

RESPONSE TO REVIEWER #1

Comment#1: Reference number 21 is omitted in the content.

Response to comment#1: This reference was omitted in error when we were rearranging the bibliography. Instead of progressing from Bibliography listed 20 to 21, we erroneously repeated 20. This omission has now been addressed in the manuscript line 123.

Comment#2: Line 475 reference using the number like in content.

Response to comment#2: This was an error, Bugembe et al 2021 is listed 64 in the manuscript’s Bibliography. This omission has now been corrected in the manuscript line 489. 

Comment#3: Line 486 spelling of beginning should be corrected.

Response to comment#3: The misspelling of the word “beginning” was a typographical error. We have corrected it as indicated in the manuscript line 500. 

RESPONSE TO REVIEWER #2

Comment#1: The author has reference 21 which was not cited anywhere in the article.

Response to comment#1: This reference was omitted in error when we were rearranging the bibliography. Instead of progressing from Bibliography listed 20 to 21, we erroneously repeated 20. This omission has now been addressed in the manuscript line 123. 

Comment#2: Line 475, the author used Vancouver referencing throughout the document, however, on this line they did not do so.

Response to comment#2: This was an error, Bugembe et al 2021 is listed 64 in the manuscript’s Bibliography. This omission has now been corrected in the manuscript line 489. 

Comment#3: Line 486 beginning, there’s a typo error

Response to comment#3: The misspelling of the word “beginning” was a typographical error. We have corrected it as indicated in the manuscript line 500.

---

## [Editor Report · Decision Letter 1]

7 Dec 2022

Retrospective in silico mutation profiling of SARS-CoV-2 structural proteins circulating in Uganda by July 2021: towards refinement of COVID-19 disease vaccines, diagnostics, and therapeutics

PONE-D-22-27027R1

Dear Dr. Prof Odongo,

We’re pleased to inform you that your manuscript has been judged scientifically suitable for publication and will be formally accepted for publication once it meets all outstanding technical requirements.

Kind regards,

Maemu Petronella Gededzha, Ph.D

Academic Editor

PLOS ONE
---

## [Editor Report · Acceptance letter]

14 Dec 2022

PONE-D-22-27027R1 

Retrospective in *silico
* mutation profiling of SARS-CoV-2 structural proteins circulating in Uganda by July 2021: towards refinement of COVID-19 disease vaccines, diagnostics, and therapeutics 

Dear Dr. Odongo:

I'm pleased to inform you that your manuscript has been deemed suitable for publication in PLOS ONE. Congratulations! Your manuscript is now with our production department. 

Kind regards, 

on behalf of

Dr. Maemu Petronella Gededzha 

Academic Editor

PLOS ONE